# *De novo* assembled mitogenome analysis of *Trichuris trichiura* from Korean individuals using nanopore-based long-read sequencing technology

James Owen Delaluna[1☉], Heekyoung Kang[1☉], Yuan Yi Chang[1,2], MinJi Kim[1,2], Min-Ho Choi[1], Jun Kim[3]*, Hyun Beom Song[1,2]*

1 Department of Tropical Medicine and Parasitology and Institute of Endemic Diseases, Seoul National University College of Medicine, Seoul, Republic of Korea, 2 Department of Biomedical Sciences, Seoul National University College of Medicine, Seoul, Republic of Korea, 3 Department of Convergent Bioscience and Informatics, College of Bioscience and Biotechnology, Chungnam National University, Daejeon, Republic of Korea

☉ These authors contributed equally to this work.
* junkim@cnu.ac.kr (JK); hbsong@snu.ac.kr (HBS)

**Data Availability Statement:** The data is available in GenBank with Accession numbers: ON646012, ON711246, and ON682760.

## Abstract

Knowledge about mitogenomes has been proven to be essential in human parasite diagnostics and understanding of their diversity. However, the lack of substantial data for comparative analysis is still a challenge in *Trichuris trichiura* research. To provide high quality mitogenomes, we utilized long-read sequencing technology of Oxford Nanopore Technologies (ONT) to better resolve repetitive regions and to construct *de novo* mitogenome assembly minimizing reference biases. In this study, we got three *de novo* assembled mitogenomes of *T. trichiura* isolated from Korean individuals. These circular complete mitogenomes of *T. trichiura* are 14,508 bp, 14,441 bp, and 14,440 bp in length. A total of 37 predicted genes were identified consisting of 13 protein-coding genes (PCGs), 22 transfer RNA (tRNAs) genes, two ribosomal RNA (rRNA) genes (*rrnS* and *rrnL*), and two non-coding regions. Interestingly, the assembled mitogenome has up to six times longer AT-rich regions than previous reference sequences, thus proving the advantage of long-read sequencing in resolving unreported non-coding regions. Furthermore, variant detection and phylogenetic analysis using concatenated protein coding genes, c*ox1*, *rrnL*, and *nd1* genes confirmed the distinct molecular identity of this newly assembled mitogenome while at the same time showing high genetic relationship with sequences from China or Tanzania. Our study provided a new set of reference mitogenome with better contiguity and resolved repetitive regions that could be used for meaningful phylogenetic analysis to further understand disease transmission and parasite biology.

## Author summary

Human trichuriasis, a neglected tropical disease caused by human whipworm *Trichuris trichiura* remains persistent in South Korea. Despite its medical importance, genomic

**Funding:** This work was supported by grant no 04-2020-0410 from the SNUH Research Fund (HBS), Creative-Pioneering Researchers Program through Seoul National University (SNU) (800-20210097 (HBS)), and the National Research Foundation of Korea (NRF) grant funded by the Korea government (MSIT) (NRF-2019R1C1C1007610 (HBS)). The funders had no role in study design, data collection and analysis, decision to publish, or preparation of the manuscript.

**Competing interests:** The authors have declared that no competing interests exist.

data about their mitochondrial DNA is scarce. In this study, we used the long-read sequencing technology of Oxford Nanopore Technologies to provide high-quality complete mitogenomes of three *T. trichiura* isolated from Korean individuals. Interestingly, our assembled mitogenomes produced up to six times longer AT-rich regions that were not reported by previous reference mitogenome sequence proving the advantage of long-read sequencing over the short-read sequencing technologies. Also, comparative analysis through variant detection and phylogenetics confirmed the distinction of our newly assembled mitogenomes over the existing references in the database. Provision of these mitogenome information is fundamental in identifying genetic markers leading to a more reliable and precise helminth diagnostics.

## Introduction

Human whipworm infection caused by *Trichuris trichiura*, is a common parasitic health problem that is categorized as Neglected Tropical Disease. Together with other parasites such as roundworm (*Ascaris lumbricoides*) and hookworm (*Ancylostoma* and *Necator* spp.), they are considered as the triad of soil-transmitted helminth (STH) infections that infest less privileged communities with poor sanitation and hygienic practice [1]. In 2015, the World Health Organization data showed that more than 1.5 billion people are affected by STH infections [2], whereas whipworm infection alone affects 477 million people worldwide [3,4]. In South Korea, the prevalence of soil-transmitted helminthiasis was higher than 60% in 1960s, but has been curved down to less than 1% prevalence since 1992 [5] and they are considered to be close to elimination. However, unlike ascariasis and hookworm infection, the trichuriasis is relatively persistent with prevalence as low as 0.4% [6].

Like the most helminths, whipworm shows high host specificity in parasitism [7,8]. The specific species, *Trichuris trichiura*, can infect human, while *T. suis* and *T. vulpis* can infect pigs and dogs, respectively [9–11]. Humans acquire the whipworm infection by ingesting soil or food contaminated with embryonated eggs that were released from adult worm of *Trichuris trichiura* in human intestine [12]. However, there is evidence showing whipworm infection across species [13,14]. Therefore, it is worthwhile to perform genetic analysis on whipworms isolated from human in the areas where human to human transmission is less likely [15].

More importantly, although these human whipworms have been studied for a long time, still there is scarcity of information about its mitochondrial genome (mitogenome). Currently, the only reference *T. trichiura* complete mitogenome available in the NCBI database is from China (GU385218) [10], Uganda (KT449826.1) [16], and another unpublished mitogenome from a group of Japanese researchers (AP017704.1). There are no records of *T. trichiura* mitogenome of Korean origin. Availability of complete mitogenome information of a specific human parasite is fundamental to parasitology research and provides further insights to its diagnostics, drug resistant strain identification, disease transmission, and phylogeographic and phylogenetic relationships [17].

In this study, we applied the long-read sequencing technology of Oxford Nanopore Technologies (ONT) utilizing third generation sequencing technology (TGS) to sequence complete mitogenomes of *T. trichiura*. Currently available *T. trichiura* reference mitogenomes were sequenced using next generation short-read sequencing (NGS) [10]. While NGS utilize 25–250 nucleotide reads [18], TGS utilize even >100-kb long reads [19] with the capability to better span the previously unknown regions of the genome, thus a single mitogenome can be covered even by a single read [20]. Among the TGS platforms, ONT offers a very cost effective

approach in sequencing non-model organisms [21,22] by using its single use adapter (Flongle) that can be mounted in a portable sequencing platform (MinION) while maintaining the same sequencing integrity [23,24]. Using the ONT long-read sequencing, we provided *de novo* assembled complete mitogenomes of three *T. trichiura* isolated from Korean individuals.

## Materials and methods

### Parasite collection and DNA extraction

The parasite samples were collected from hospitals in South Korea from 2020–2021 and referred for morphologically diagnosis to the department of Tropical Medicine and Parasitology, Seoul National University College of Medicine. As anonymized and residual materials were used, formal consents were not obtained, and they were considered exempt from requiring research ethics approval by Institutional Review Board of Seoul National University Hospital.

The samples were washed in DEPC water to remove ethanol, frozen in liquid nitrogen, then homogenized with stainless steel beads in TissueLyser LT (QIAGEN) at 50 Hz, 3 cycles, 2 minutes per cycle. Then, genomic DNA was extracted using DNeasy tissue and a blood kit (QIAGEN) according to the manufacturer's instructions.

### Oxford Nanopore sequencing

**MinION library preparation.** Extracted DNA of adult *T. trichiura* was prepared using the ONT MinION sequencing Kit (SQK-LSK109) with slight modifications on the manufacturer's protocol. Briefly, DNA repair and tailing were performed using NEBNext FFPE DNA repair Mix (cat. no. M6630S) and NEBNext Ultra II End repair / dA-tailing Module reagents (cat. no. E7546S) (New England Biolabs) by adding 24 μL of DNA samples with the reagents to make 30 μL reaction and then incubated at 20˚C for 5 min and 65˚C for 5 min. The repaired/ end-prepped DNA was then cleaned up using AMPure XP beads (cat. no. A63880, Beckman Coulter Inc.) following manufacturer's protocol. Samples were kept on the magnetic rack and washed twice with 200 μL freshly prepared 70% ethanol in nuclease-free water. Then 30 μL DNA sample was mixed with adapter reagents to produce 50 μL reaction and incubated at room temperature for 10 min followed by final clean up with AMPure XP product. Beads were washed off using Fragment buffer in the ONT kit before resuspending with 7 μL Elution buffer at room temperature for 10 min followed by additional incubation at 37˚C to optimize recovery. Eluted sample was quantified using the Qubit fluorometer and adjusted to ensure that sample concentration is within the recommendation (3–20 fmol, purity 1.8) before loading onto the flow cell. Lastly, flush buffer (117 μL) and flush tether (3 μL) were loaded in a Flongle flow cell in preparation for the sequencing. The final genomic library was mixed with sequencing buffer and loading beads to make a total of 30 μL reaction and ready for sequencing.

**MinION sequencing.** Long-read sequencing was performed using Flongle flow cells inserted in the MinION portable sequencer (Oxford Nanopore Technologies) connected to the computer. MinKNOW software (v22.03.5) was used to assess flow cell quality and monitor pore activity. Prepared genomic library was then loaded to the flow cell and sequencing was run for 24-h period following manufacturer's recommendation. Only the FASTQ output files (raw reads) were used in assembly steps.

### Genome assembly, polishing, and circularization of mitogenomes

**Preparation of mitochondrial ONT reads.** Raw reads that specifically span the mitochondrial genome of the reference whole genome were utilized for the assembly to remove any

non-mitochondrial genomic sequence in our long-read sequencing data. To obtain these mito-chondrial reads, firstly, the mitochondrial region of the reference whole genome (GCA_000613005.1_TTRE2.1) was annotated using a previously published mitogenome (GU385218) downloaded from the NCBI genome database. Annotation was performed by aligning GU385218 to GCA_000613005.1_TTRE2.1 using *nucmer* and *show-tilling* from the MUMmer package [25,26]. This annotated mitogenome in the reference genome serves as a template to characterize raw mitochondrial ONT reads. Secondly, our ONT raw whole-genome reads were mapped to the annotated reference mitogenome using minimap2 [27], and alignments were sorted and indexed using SAMtools [28]. Thirdly, the ONT raw reads mapped to the reference mitogenome were extracted and categorized as mitochondrial reads in our samples using SAMtools and seqtk.

**_De novo_ mitogenome assembly.**    We performed reference-guided mitochondrial read preparation and *de novo* assembly of the reads based on the method introduced by Schneeber-ger et al. [29]: raw reads were initially mapped to a reference genome, the spanning reads were extracted as mitochondrial reads, and these mitochondrial reads were used to create a com-plete mitochondrial contig. Briefly, extracted ONT reads having >20× coverage were assem-bled using Canu [30] with the parameter "*14k*" as the estimated *Trichuris trichiura* mitogenome size. For our three assembled mitogenome based on ONT long reads, each mito-genome sequence was polished using the alignment information of their corresponding raw reads by minimap2 and RACON [31]. This polishing step is repeated once again. The polished mitogenomes are then circularized after removing overhang regions using the Geneious prime (version 2022.2.2). These circularized mitogenomes were rearranged in relation to the starting nucleotides of the reference mitogenomes using *nucmer* and *show-coords* in the MUMmer package. Visualization of the circularized mitogenome and its features was conducted in the CGView tool (cgview.ca). All scripts for each step are compiled in S1 File.

**Genome annotation and analysis.**    Assembled mitogenomes were annotated in terms of variants and genes using Geneious prime (version 2022.2.2). Variants in the mitogenomes were detected in the galaxy platform (usegalaxy.org). First, *bcftools mpileup* and *bcftools calling* were used to produce a VCF file, then variants were annotated using SnpEff [32]. Variants called were tabulated and used to create a Venn diagram (Venny.com). The copy number vari-ation (CNV) of AT-rich region were detected using Tandem Repeat Finder (tandem.bu.edu). Pairwise comparison was performed in Geneious prime whereas concatenated protein coding genes (PCGs), *cox1*, and *rrnS* regions of each mitogenome were extracted and aligned with the corresponding reference sequences to report variability and create a heatmap using R studio (version R.4.1.2). Also, all mitogenomes were aligned to each other and compared using *dna-diff* in the MUMmer package [25,26]. Then, principal component analysis (PCA) was per-formed using average nucleotide identity (ANI) of each pair of mitogenomes. Results were tabulated to produce a PCA graph in R studio (version R.4.1.2). All scripts for each step are compiled in S1 File.

**Phylogenetic analysis.**    Phylogenetic analysis was performed using Maximum Likelihood (ML) method. Our assembled mitogenomes were compared with *cox1* sequences, *rrnL* sequences, and *nd1* sequences of *Trichuris* sp. from non-human hosts and *T. trichiura* from other countries. The highly conserved *cox1* region was chosen to confirm the clustering between our assembled mitogenomes and *Trichuris* spp. while *Trichinella spiralis* is used as an outgroup. Lists of sequences used in this analysis are in S2 File. Datasets were aligned using ClustalW using default parameters. Then aligned sequences were trimmed manually in MEGA X to remove unaligned codons and nucleotides. The ML tree was generated using iQTree [33]. Best-fit substitution model was determined using the jModelTest [34] on CIPRES [35]. For the *cox1* and *nd1* dataset, the mtREV + G model was chosen [36], while HKY G+I

model was chosen for the *rrnL* dataset. Bootstrap support for topology was set to (mini-heuristic option) 1000 replications. Nucleotide substitution rates were also shown in the consensus phylogenetic tree. All trees were saved in newick format and visualized in Figtree v1.4.4 (http://tree.bio.ed.ac.uk/software/figtree/).

## Results

### Characteristics of *de novo* assembled mitochondrial genomes using a long-read sequencing approach

We sequenced genomic DNA of three *Trichuris trichiura* samples from Korean individuals and assembled the sequencing reads into three mitogenomes (GenBank Accession numbers: TTK1—ON646012, TTK2—ON711246, TTK3—ON682760). Initially, the analysis was intended for whole genome assembly but due to the low sequencing depths of the whole genome data, we shifted and focused on the mitochondrial genome analysis, as the mitochondrial genome has much higher copy number than that of the nuclear genome. Nearly 15,128 high-quality long reads were *de novo* assembled into a 14.5-kb sequence with an N50 of up to 4234 bp. All assembled complete mitogenomes have raw read sequencing depths of 74×, 30×, and 52×, respectively (S1 Table) (raw data are available in the Sequence Read Archive under the accession number PRJNA823754). Among the three mitogenomes, TTK1 exhibited the longest mitogenome length (14,508 bp) and the best resolved AT-rich region (see below) (Fig 1). All of our mitogenomes were composed of ~69.0% A + T bases, which is slightly higher than those of previously published reference mitogenomes (~60.8%). This AT-bias can be

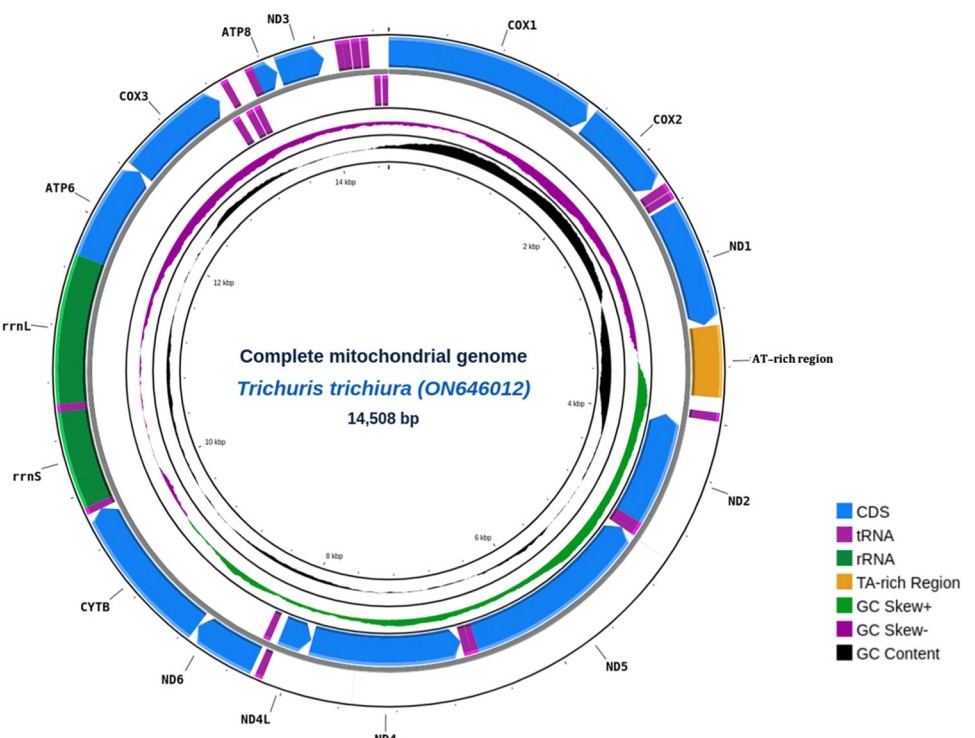

**Fig 1. *Trichuris trichiura* complete mitogenome circular map showing annotated features.** Schematic representation of the mitogenome including 13 Protein coding genes, 2 rRNAs and 22 tRNAs. AT-rich region that span 6× longer than reference mitogenomes lies between *nd1* and *nd2*. Inner most ring shows the GC content and GC skew of each regions.

**Table 1. Features of *de novo* assembled *Trichuris trichiura* mitogenomes from Korean patients and published reference mitogenomes.**

| Mitogenome | Sequencing technology | Length (bp) | AT repeat length (bp) | AT rich region (%) | AT repeats (Copy Number Variation) | GC (%) |
|---|---|---|---|---|---|---|
| TTK1 *Trichuris trichiura*-Korea (ON646012) | ONT long read sequencing (9,173 reads) | 14,508 | 509 | 3.51 | 255.0 | 31.0 |
| TTK2 *Trichuris trichiura*-Korea (ON711246) | ONT long read sequencing (15,128 reads) | 14,441 | 452 | 3.13 | 226.5 | 39.4 |
| TTK3 *Trichuris trichiura*-Korea (ON682760) | ONT long read sequencing (5,651 reads) | 14,440 | 447 | 3.09 | 224.0 | 39.4 |
| TTJP *Trichuris trichiura*-Japan (AP017704) | Illumina short read sequencing | 14,092 | 87 | 0.62 | 44.0 | 31.9 |
| TTCN *Trichuris trichiura*-China (GU385218) | Long-range PCR Sequencing-unspecified | 14,046 | 53 | 0.38 | 27.0 | 39.2 |
| TTUG *Trichuris trichiura*-Uganda (KT449826) | Sanger sequencing | 14,079 | 26 | 0.18 | 13.5 | 30.7 |

explained by an unresolved AT-repeat cluster in the previously published mitogenomes, as they were assembled using short-read sequencing technologies. Indeed, the length of the AT-repeat cluster in our mitogenome (255 copies of AT sequences) was six times longer than any of the previous reference mitogenomes (Table 1). The raw-read depth of the AT-repeat cluster of our mitogenome was comparable to that of the other mitogenome sequences (ratio 1.09; 3.09~3.51% while references has 0.18~0.62%), which suggests that the AT-repeat cluster was assembled with highest reliability. This emphasize the superior capability of long-read sequencing in resolving a long span of AT repeats in the genome compared to that of short-reads sequencing.

## Sequence annotation and variant identification compared to *T. trichiura* reference sequences

We annotated genes in our mitogenome based on nucleotide similarity with the gene sequences of the reference. Consistent with all previously published mitogenomes, 37 predicted genes and two non-coding regions were identified that consist of 13 PCGs (*cox1–3*, *nad1–6*, *nad4L*, *cytb*, *atp6*, and *atp8*), 22 tRNA genes, and two rRNA genes (*rrnS* and *rrnL*). The gene order and length of the PCGs are the same with that of the reference mitogenomes, but the nucleotide positions varied due to the difference in the length of AT-repeat cluster (S2 Table).

Genetic variants were called between the assembled and reference mitogenomes. When setting the reference mitogenome with TTCN, the most recently published mitogenome of *T. trichiura*, TTK1 had higher number of variants in all genes than TTK2 and TTK3 with variants ranging from 4.7%–9.2% in each PCG. In particular, *nd2*, *nd3*, *nd4* and *nd4L* regions showed hypervariability (5.26%~9.15%) while *cox1* was found to be the least variable in TTK1 (Table 2).

In addition, we also assessed possible impacts of these genetic variants using SnpEff. Synonymous and non-synonymous genetic variants differ between our assembled mitogenomes. The overall transition and transversion ratio ranges from 14.46–18.80 while the missense and silent mutation ratio was 0.42–0.67 (S3 Table).

**Table 2. Variants called in each gene region between our mitogenomes and the reference sequence.** *gene length of each PCGs in the TTCN mitogenome | ** variant ratio = gene variant count / gene length (bp).

| Gene region | *Length (bp) | Variants (counts \| ratio**) | | |
|---|---|---|---|---|
| | | TTK1/TTCN | TTK2/TTCN | TTK3/TTCN |
| *cox1* | 1545 | 73 \| 0.047 | 20 \| 0.013 | 22 \| 0.014 |
| *cox2* | 675 | 44 \| 0.065 | 9 \| 0.013 | 7 \| 0.010 |
| *nd1* | 900 | 43 \| 0.048 | 10 \| 0.011 | 13 \| 0.014 |
| *nd2* | 885 | 81 \| 0.092 | 10 \| 0.011 | 15 \| 0.017 |
| *nd5* | 1548 | 126 \| 0.081 | 22 \| 0.014 | 27 \| 0.017 |
| *nd4* | 1212 | 108 \| 0.089 | 27 \| 0.022 | 22 \| 0.018 |
| *nd4L* | 258 | 20 \| 0.078 | 10 \| 0.039 | 8 \| 0.031 |
| *nd6* | 477 | 38 \| 0.080 | 7 \| 0.015 | 6 \| 0.013 |
| *cytb* | 1107 | 75 \| 0.068 | 16 \| 0.014 | 19 \| 0.017 |
| *atp6* | 828 | 67 \| 0.081 | 21 \| 0.025 | 20 \| 0.024 |
| *cox3* | 774 | 66 \| 0.085 | 21 \| 0.027 | 19 \| 0.025 |
| *atp8* | 165 | 14 \| 0.085 | 4 \| 0.024 | 3 \| 0.018 |
| *nd3* | 342 | 18 \| 0.053 | 6 \| 0.018 | 11 \| 0.032 |

With three mitogenomes we obtained and three mitogenomes from previous studies, we tried to figure out relationships in all six mitogenomes in terms of pairwise variation analysis using concatenated PCGs, *cox1*, and *rrnL* sequences (Fig 2). Concatenated PCGs, *cox1* and *rrnL* sequences revealed similar patterns of sequence differences in the six mitogenomes, but not perfectly overlapped (S3 Table).

Impressively, we found that *T. trichiura* mitogenomes are extremely divergent, as some mitogenomes exhibited ~18% sequence differences even in the most conserved *cox1* gene (S4 Table). Specifically, the Ugandan mitogenome was most different from any other mitogenomes, and this feature was not dependent on sequencing technologies, as pairs of short-read sequencing-based mitogenomes also exhibited 16.8%–20.7% of sequence differences. TTK1 is similar to the mitogenome from Japan, while both TTK2 and TTK3 were more similar to the mitogenome from China (Fig 3).

## Phylogenetics analysis based on the mitogenome

To confirm the molecular identity of our assembled sequences and its relationship to previously reported sequences, phylogenetic analyses were performed using three mitochondrial

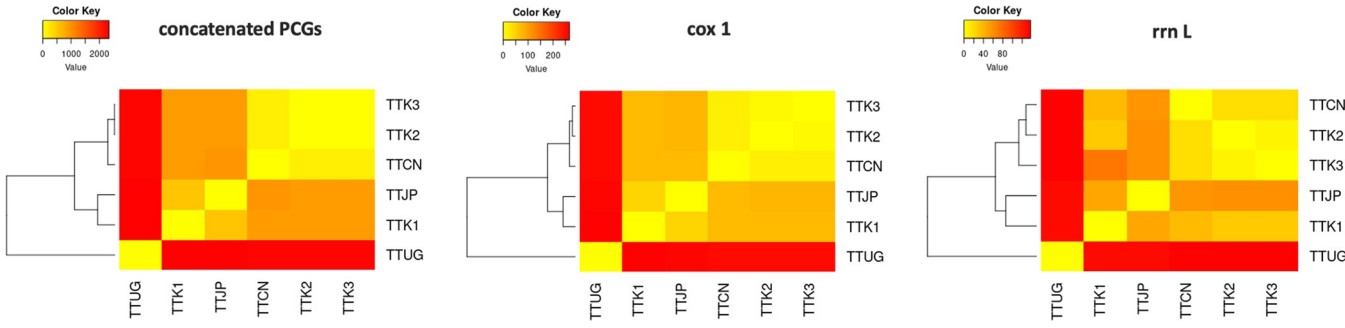

**Fig 2. Heatmap of mitogenome variability.** Pairwise results of concatenated PCGs, *cox1* region, and *rrn L* region of our assembled mitogenomes (TTK1-K3) and of Japan (TTJP), China (TTCN), and Uganda (TTUG) reference mitogenomes.

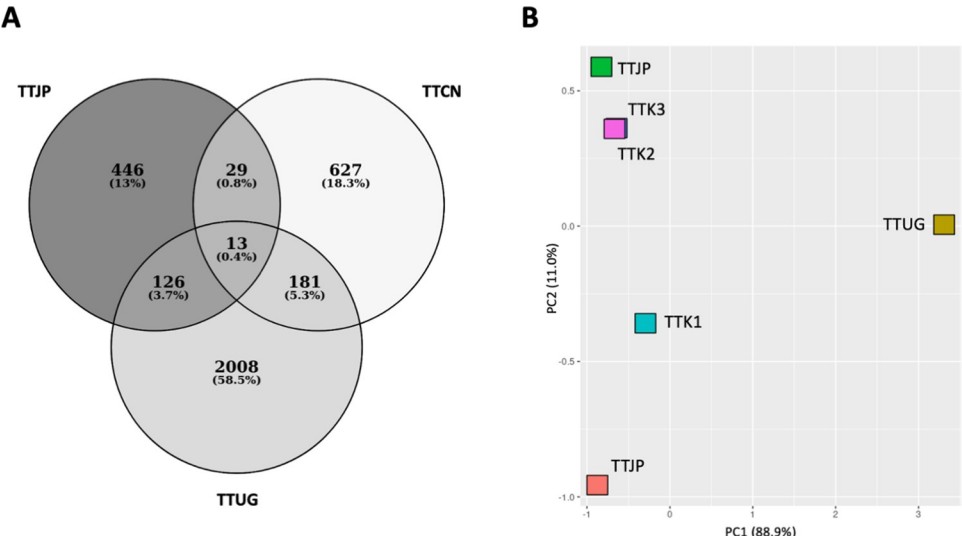

**Fig 3. Genetic similarities between our assembled mitogenomes and the reference mitogenomes.** (A) Venn diagrams showing the unique and shared single nucleotide variations (SNVs) of TTK1 mitogenome in relation to each reference mitogenome. (B) Principal component analysis (PCA) using average nucleotide identity of each mitogenome.

markers namely: *cox1*, *rrnL*, and *nd1* sequences. Consensus phylogenetic trees constructed using ML method exhibited similar tree topology, which also remain consistent throughout the different mitochondrial markers used.

Two major clades corresponding to *T. trichiura* and *T. suis* were observed in the tree produced using *cox1* genes of whipworms collected from pigs, non-human primates and humans from different geographic regions (Fig 4). *T. suis* population consisting of worms from China, Uganda, Denmark, and Spain formed its own separate subclade while in the same major clade with whipworms derived from Old world monkeys; *C. g. kikuyensis* (mantled guereza), *P. ursinus* (chacma baboon) and *C. sabaeus* (green monkey). Leaf monkeys (*T. francoisi*) formed two separate sister relationship within the *T. trichiura* clade. The first group clustered with most of the human-derived *Trichuris* while the second group formed separate from the rest of the *Trichuris* sp. Similarly, Japanese macaque (*M. fuscata*) formed its own subclade under the *T. trichiura* clade while in a sister relationship with a subclade composed of *T. trichiura* sequences from human in Uganda together with most of the non-human primate derived worms from Chacma baboons (*P. hamadrayas*), Barbary ape (*M. sylvanus*) and *Papio* species.

In addition, a separate subclade of whipworms from humans showed three separate monophyletic groups where Group 1 consists of sequences from Chinese patients together with our assembled sequences TTK2 and TTK3. Group 2 consists of our assembled sequence TTK1, a sequence from a human in Tanzania, and a NHP derived sequence from a Guinea baboon (*P. papio*) while Group 3 is composed of the rest of the *T. trichiura* sequences from other geographic locations namely; Japan, Ecuador, Honduras, Cameroon and Tanzania (Fig 4).

To provide a better resolution to the relationship between human-derived and NHP-derived *Trichuris*, we use took advantage of the variable nature of *rrnL* sequences to detect intraspecies relationship (Fig 5). The results of *rrnL* phylogenetic analysis confirmed the relationship observed using *cox1* gene markers. However, in the *rrnL* tree, the Guinea baboon (*P. papio*) sequence did not form a sister relationship as previously observed in *cox1* tree, rather it was positioned in between the monophyletic group of China and our TTK2 and TTK3 sequences and the group of TTK1 other human derived sequences except for Uganda and

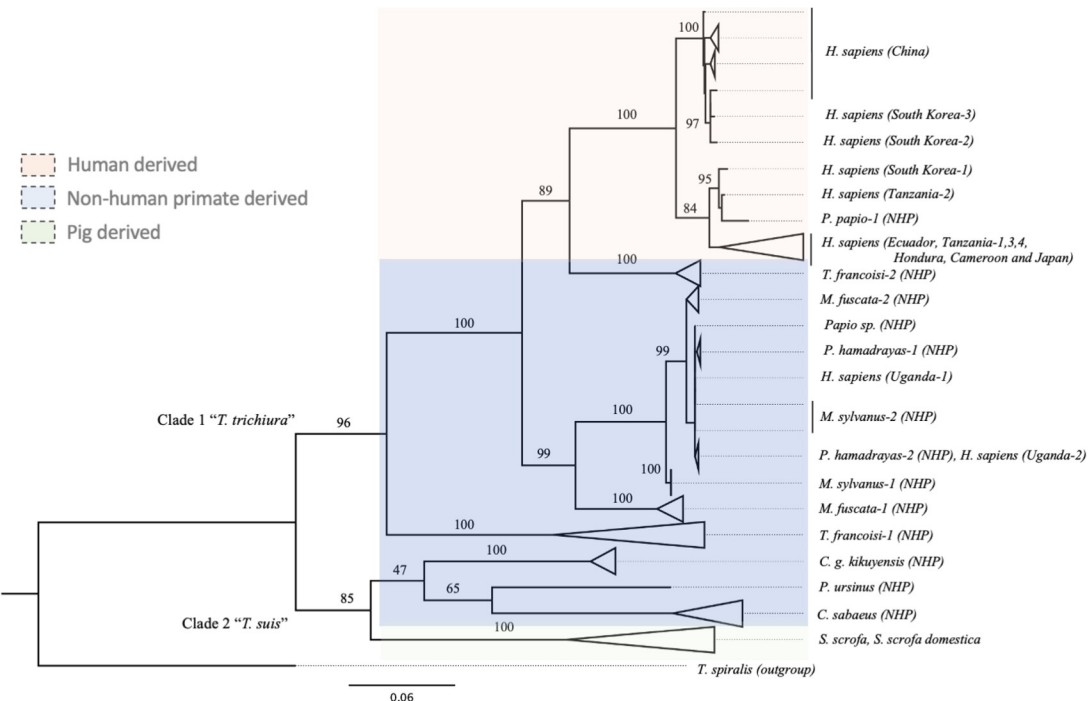

**Fig 4. Maximum likelihood tree showing the amount of genetic variability that has occurred between Trichuris species extracted from pigs, human and non-human primates.** Analysis is based on *cox1* genes of our assembled sequence together with sequences reported and used by Cavallero et al., 2019 [37] and Doyle et al., 2022 [38]. A total of 106 sequences were used including the outgroup. Bootstrap values of branches with less than 85 value were not presented. Triangles represent collapsed cluster of group of taxa or clade within the tree.

China. Consistently, human whipworms from Uganda clustered with NHP-derived worms from *P. hamadrayas*, *M. sylvanus*, and Papio sp. monkeys. Meanwhile, some of the sequences of *Trichuris* from *T. francoisi* and *M. fuscata* formed their separate cluster, respectively.

Then, hypervariable *nd1* sequence was used to produce a detailed tree of whipworms from human excluding sequences from human in Uganda that were consistently clustered with NHP derived sequences. The *nd1* tree confirmed the phylogenetic relationship of human and NHP derived whipworm sequences using *cox1* and *rrnL* gene markers (Fig 6). *P. papio* formed a sister relationship with our assembled sequence TTK1 and sequence form human in Tanzania similar to Group 1 in *cox1* results. The results also emphasized the geographical distribution of human whipworms where China and Korea (TTK2 and TTK3) formed its own cluster like Group 2 in *cox 1* results, which is separate from Ecuador, Honduras, Cameron and Japan (Group 3). In addition, Group 3 in *cox 1* tree could be divided into 2 subgroups that hold Japan sequence and Tanzania sequence in each subgroup while Ecuador, Honduras, Cameroon sequences are distributed in both subgroups, not segregated.

## Discussion

Knowledge about mitogenomes has been proven essential in human parasite diagnostics, however the lack of substantial data for comparison is still a challenge in *T. trichiura* research. Here we generated three complete mitogenomes of *T. trichiura* isolated from Korean patients using ONT long-read sequencing technology. To our knowledge, we report the first complete mitogenome of *Trichuris trichiura* from Korea. The main feature of our assembled mitogenome is the longer span of the AT-repeat cluster that has not been perfectly resolved due to the limitations of short-read sequencing technique.

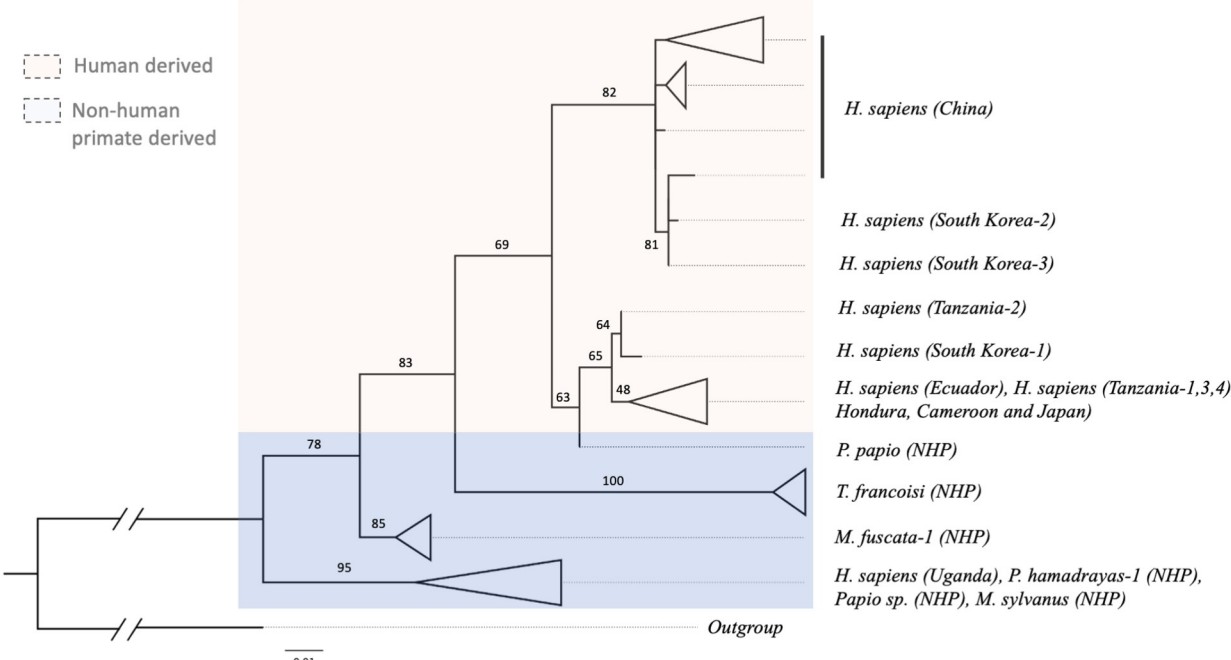

**Fig 5. Maximum likelihood tree showing the amount of genetic variability that has occurred between *Trichuris* species extracted from human and non-human primates.** Analysis is based on *rrnL* genes of our assembled sequence together with sequences reported and used by Cavallero et al., 2019 [37] and Doyle et al., 2022 [38]. A total of 98 sequences were used including the outgroup. Bootstrap values of branches with less than 60 value were not presented. Triangles represent collapsed cluster of group of taxa or clade within the tree.

Our findings emphasize the advantage of TGS long-read sequencing (>10 kb read-length) in terms of resolving repeats over the second generation or NGS short-read sequencing technologies (<500 bp). Illumina sequencing uses short reads by synthesis while ONT sequencing directly allow the DNA strands to pass a nanopore and detect the changes in ionic current which correspond to specific nucleotide [23]. Currently, the major issue of using TGS is the higher error rate than that of NGS given the single-molecule sequencing chemistry of TGS. However, this is compensated by increasing the sequencing depth and the use of updated platforms with reliable quality control measures [17]. Illumina deep sequencing has the advantage of better variant detection but is limited to incomplete reconstruction of the genomic information while ONT offers portability and cost effective construction of complete genomic information despite its higher error rate [39]. Among the two TGS platforms, PacBio sequencing can produce higher base-level quality of reads with lower error rate compared to ONT or even NGS [40,41]. However, ONT is gaining popularity through its reliability in producing slightly longer mappable reads at a lower cost (1,000–2,000 USD) [42]. Availability of cheaper alternatives such as ONT sequencing will open more opportunities for non-model species to be sequenced, thus lessen the scarcity of genomic information in the database. Thus in this study we opted to use ONT to report the first *T. trichiura* complete mitogenome from Korea.

Just like any other eukaryotic organisms, nematodes follow a strict maternal inheritance in their mitochondrial genomes [43]. The role and interaction of mitochondrial and nuclear genomes are fundamental in ensuring efficient ATP synthesis by coordinating protein function and RNA production [44]. Interestingly, it was shown that the relevance of mitochondrial genomes to the nuclear genomes lies in the fact that mitonuclear interaction has been observed to have effects on the nuclear genome's physiology and direction of evolution [45]. In fact, similar tree topologies have been observed for Nematoda phylogenetic analysis using

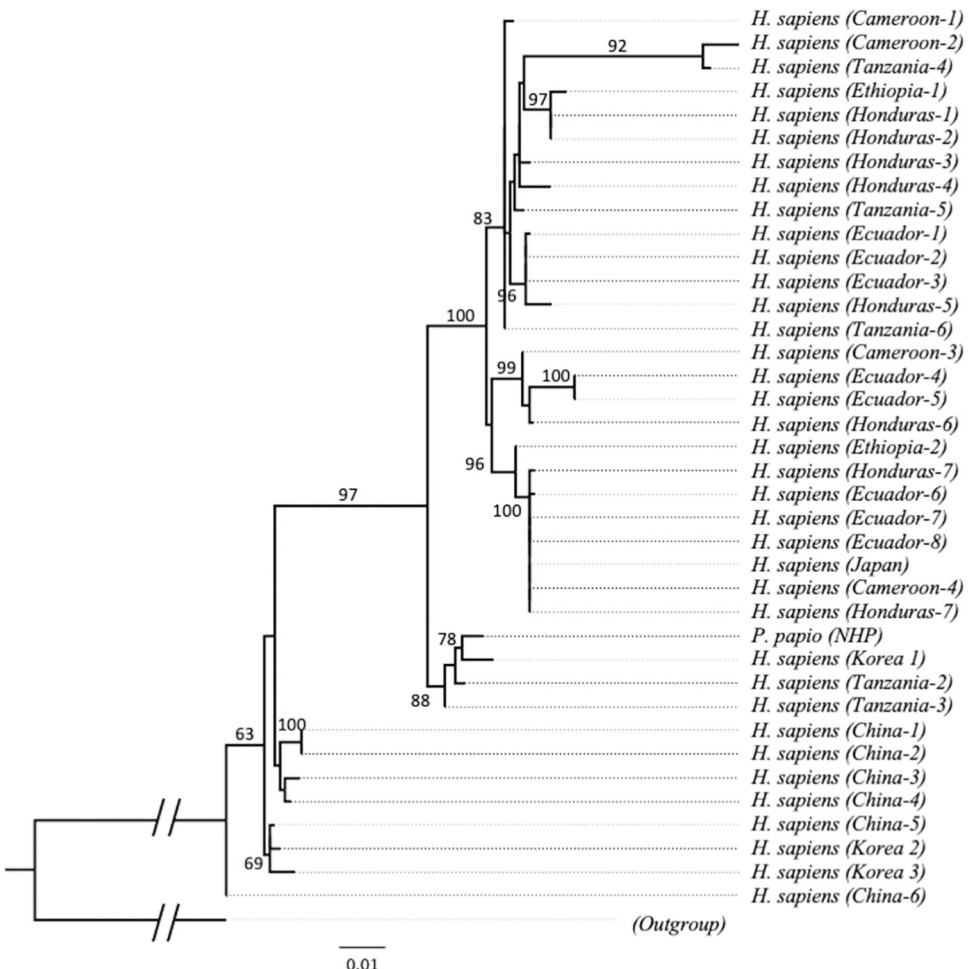

**Fig 6. Maximum likelihood tree showing the amount of genetic variability between Trichuris species extracted from human and non-human primates.** Analysis is based on *nd1* genes of our assembled sequence together with sequences reported and used by Cavallero et al., 2019 [37] and Doyle et al., 2022 [38]. A total of 39 sequences were used including the outgroup. Bootstrap values of branches with less than 60 value were not presented. Triangles represent collapsed cluster of group of taxa or clade within the tree.

mitochondrial and nuclear genomes [46,47]. Thus, mitochondrial genomes may serve as a complementary solution in instances where nuclear genomes fail to provide better species resolution or when morphologic and genetic data are contradicting [48].

Compared to the available reference complete mitogenome sequences of *T. trichiura* from humans, there is a substantial extension in the length of our assembled mitogenome of *T. trichiura*, mostly attributing to the previously unresolved AT repeats in the non-coding region. Mitogenomes are known to have high mutation rates due to its susceptibility to deletions particularly in non-coding regions with high repeats [49]. These highly clustered repeats in noncoding sections of mitogenomes have been suggested to be negatively associated with mammalian longevity [50]. These variations in the copy number of mitochondrial AT repeats have been used to measure intraspecific genetic variability [51]. In the setting of infectious diseases such as schistosomiasis [52] and tuberculosis [53], the repeats serve as genetic markers in resolving intraspecific differences, super-infection, and mixed infections. They may also play an important role in DNA replication and transcription as initiation sites for polymerase

binding and termination [54]. Thus, it is also important to correctly assemble and resolve AT repeats in the non-coding region, which could be accomplished by sequencing with ONT in this study.

Pairwise comparison of our mitogenomes with reference sequences revealed relatively conserved PCGs and highly variable regions. Consistent previous studies [46,55], we also report *cox1* as the most conserved PCGs. So far, *cox1* is widely used in mitochondrial gene analyses to further understand species diversity, diagnostics, and population variation [36]. On the other hand, NADH dehydrogenase subunit (*nad*) genes showed highest variability among PCGs. Between *cox1* and high-variable regions, the latter is more recommended for species prospecting [55]. These variable regions can even be used to differentiate between isolates of the same species collected from one host. That is, if multiple whipworms were extracted from one patient, we can amplify these regions from each worm to potentially detect if they came from the same maternal origin.

Phylogenetic analysis results provided evidence that our assembled mitogenomes were indeed *T. trichiura* sequences yet distinct from the reference sequences in the database. Using the highly conserved *cox1* gene as mitogenome marker we report separate clades of *T. trichiura* and *T. suis* and all our sequences belong to *T. trichiura*. This distinction is consistent with previous reports [14,16,46]. Within the clades of *T. trichiura*, distinction between human and NHP-derived whipworms were clearly demonstrated by phylogenetic analysis using *cox1* and *rrnL* genes, which is consistently shown by several studies [9,37,46,56].

However, some sequences support the transmission of *Trichuris* species shared between human and non-human primates [16]. *T. trichiura* from Uganda was clustered with a few *Trichuris* from non-human primates across all mitogenome markers used in this analysis, and one *Trichuris* from Guinea baboon (*P. papio*) somewhat clustered with our TTK1 and *T. trichiura* from Tanzania by *cox1* and *nd1* gene analysis. Similar finding was described in a previous study [37] which utilized *cox1*, *cob* and concatenated protein-coding genes to report the complex relationship among *Trichuris* sp. that infects primates.

Within *T. trichiura* from human, we identified 3 groups: TTK2 and TTK3 clustered with sequences from China, TTK1 clustered with some sequences from Tanzania and a sequence from Guinea baboon, and rest of the *T. trichiura* sequences from other geographic locations. By using the hypervariable region *nd1* gene, the last group could be divided into two subgroup that is not generally segregated by locations. Additional sequences from different locations are needed to clearly demonstrate intraspecies clustering. The close phylogenetic relationship between *T. trichiura* sequence from Korea and China was first reported by Hong et al., [57] where they amplified ancient DNA collected from latrines in mummified sites in Korea. Interestingly, this is the first report of a close phylogenetic relationship between *T. trichiura* sequence from Korea and Tanzania.

Since this topology is consistent even when highly variable genomic region is used for the same analysis, there is a high chance that our samples are local infections given the evidence of its significant distinction from other reference sequences from neighboring countries. Although in a previous study, Hawash et al., [16] suggested dispersal of *Trichuris* infection from Africa to Asia and TTK1 is clustered with sequence from Tanzania, other two mitogenomes are distinct from previously reported *T. trichiura* from Africa and closer to sequences from a neighboring Asian country, China.

In this study, we provided information about complete mitogenomes of *T. trichiura* isolated from Korean individuals. The cost-effectiveness and portability of using ONT in producing complete mitogenome sequences rather than the more laborious whole genome sequences is very promising for resource limited countries with rampant parasitic infections. Our *de novo* assembled mitogenomes were longer than any other references benefiting from long-read

sequencing. Furthermore, comparative analysis revealed that they were not clustered with *Trichuris* spp. isolated from non-human hosts but were clustered either with mitogenomes from China or Tanzania. Our study provided a new set of reference mitogenome with better contiguity and resolved repetitive regions that could be used for meaningful phylogenetic analysis to further understand disease transmission and parasite biology.

## Supporting information

**S1 Table. Raw reads statistics.**
(DOCX)

**S2 Table. Features of annotated protein coding genes our mitogenome and the reference sequences.**
(DOCX)

**S3 Table. Snpeff variant calling results.** *T. trichiura* China (GU385218) is used as reference genome for variant calling since it is the closest published reference mitogenome.
(DOCX)

**S4 Table. Variability of mitogenomes.** Pairwise comparison results between assembled mitogenomes and reference sequences.
(DOCX)

**S1 File. Custom Script.**
(DOCX)

**S2 File. List of sequences included in the analysis.**
(DOCX)

## Author Contributions

**Conceptualization:** Min-Ho Choi, Hyun Beom Song.

**Data curation:** James Owen Delaluna, Heekyoung Kang.

**Formal analysis:** James Owen Delaluna.

**Funding acquisition:** Hyun Beom Song.

**Investigation:** James Owen Delaluna, Heekyoung Kang, Yuan Yi Chang, MinJi Kim.

**Methodology:** James Owen Delaluna, Jun Kim.

**Resources:** Heekyoung Kang, Yuan Yi Chang, MinJi Kim, Min-Ho Choi, Hyun Beom Song.

**Software:** James Owen Delaluna, Jun Kim.

**Supervision:** Jun Kim, Hyun Beom Song.

**Validation:** Heekyoung Kang, Jun Kim.

**Visualization:** James Owen Delaluna.

**Writing – original draft:** James Owen Delaluna, Heekyoung Kang.

**Writing – review & editing:** James Owen Delaluna, Heekyoung Kang, Yuan Yi Chang, MinJi Kim, Jun Kim, Hyun Beom Song.

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
