## [Decision Letter · Decision Letter 0]

12 May 2023

Dear Dr. Song,

Thank you very much for submitting your manuscript "De novo  assembled mitogenome analysis of Trichuris trichiura from Korean individuals using nanopore-based long-read sequencing technology" for consideration at PLOS Neglected Tropical Diseases. As with all papers reviewed by the journal, your manuscript was reviewed by members of the editorial board and by several independent reviewers. The reviewers appreciated the attention to an important topic. Based on the reviews, we are likely to accept this manuscript for publication, providing that you modify the manuscript according to the review recommendations. 

Since the primary submission, the methodology was substantially improved based on editor feedback. The current reviewers both do not have concerns about the methodological approach, but the authors should consider the suggestions from reviewer 1 to include more sequences from NHP T. trichiura to address previous findings of identical strains from NHP and humans, as well as to explore results with different markers. Additionally, including versions of the R packages will facilitate repeatability of the methods in the future. 

However, the reviewers have disagreed about the novelty of the publication. While it's acknowledged that sequencing of T. trichiura mitochondrial genomes is not entirely novel, as outlined in the introduction, there is a currently a lack of knowledge about how many distinct strains there are, and their host specificity, which are both important for efforts to control infections. Providing a better quality mitochondrial genome, adding more distinct strains to the available literature, and developing a methodology to use long reads for future experiments and comparisons provides a "good resource for parasite genomics and diagnostics", as specified by reviewer 1. As such, we will consider this to be novel enough for publication in PLOS NTD.

Sincerely,

Bruce A. Rosa

Academic Editor

Cinzia Cantacessi

Section Editor

Since the primary submission, the methodology was substantially improved based on editor feedback. The current reviewers both do not have concerns about the methodological approach, but the authors should consider the suggestions from reviewer 1 to include more sequences from NHP T. trichiura to address previous findings of identical strains from NHP and humans, as well as to explore results with different markers. Additionally, including versions of the R packages will facilitate repeatability of the methods in the future. 

However, the reviewers have disagreed about the novelty of the publication. While it's acknowledged that sequencing of T. trichiura mitochondrial genomes is not entirely novel, as outlined in the introduction, there is a currently a lack of knowledge about how many distinct strains there are, and their host specificity, which are both important for efforts to control infections. Providing a better quality mitochondrial genome, adding more distinct strains to the available literature, and developing a methodology to use long reads for future experiments and comparisons provides a "good resource for parasite genomics and diagnostics", as specified by reviewer 1. As such, we will consider this to be novel enough for publication in PLOS NTD.

Reviewer's Responses to Questions

**Key Review Criteria Required for Acceptance?**

**Methods**

-Are the objectives of the study clearly articulated with a clear testable hypothesis stated?

-Is the study design appropriate to address the stated objectives?

-Is the population clearly described and appropriate for the hypothesis being tested?

-Is the sample size sufficient to ensure adequate power to address the hypothesis being tested?

-Were correct statistical analysis used to support conclusions?

-Are there concerns about ethical or regulatory requirements being met?

Reviewer #1: Yes Methods are adequate and suitable for the study

Reviewer #2: Yes

**Results**

-Does the analysis presented match the analysis plan?

-Are the results clearly and completely presented?

-Are the figures (Tables, Images) of sufficient quality for clarity?

Reviewer #1: Results are clearly presented

Reviewer #2: Yes

**Conclusions**

-Are the conclusions supported by the data presented?

-Are the limitations of analysis clearly described?

-Do the authors discuss how these data can be helpful to advance our understanding of the topic under study?

-Is public health relevance addressed?

Reviewer #1: Conclusions are supported by the results

Reviewer #2: Yes

**Editorial and Data Presentation Modifications?**

Reviewer #1: (No Response)

Reviewer #2: (No Response)

**Summary and General Comments**

Reviewer #1: In the study by Delaluna and colleagues entitled “De novo assembled mitogenome analysis of Trichuris trichiura from Korean individuals using nanopore-based long-read sequencing technology”, the authors characterized whole mitochondrial genomes of three human Trichuris, Trichuris trichiura, isolates from 3 human patients in South Korea using long read sequencing of Nanopore. The authors analyzed the mitochondrial genomes using variants characterization and phylogenetic analysis with other published mitochondrial genomes. 

The manuscript represents a good resource for parasite genomics and diagnostics since it is the first study to characterize mitochondrial genomes of human Trichuris using long read sequencing which had an advantage of characterizing the genome with high accuracy (e.g. precise estimation of AT region). The methods are suitable and the conclusions are supported by the results. Although the overall conclusion is not novel, it supports the potential existence of two species of human Trichuris. 

I have only a few suggestions. 

In Figure 6, the authors identified 3 clusters that separate Trichuris from 3 hosts, humans, NHP and pigs. There are more rrnL sequences from NHP that suggest it is not differentiable from human Trichuris. I suggest the authors to include more sequences of rrnL from NHP and make more comprehensive tree that reflects the complexity of the Trichuris phylogeny. 

Also, the authors used cox1 and rrnL when exploring the relatedness between Trichuris from different species. However, both markers are conserved markers. I suggest to also include other variable markers such as nad1 to give a phylogenetic relationship from a different perspective that may resolve a potential hidden sub-cluster of Trichuris. 

One minor comment, I encourage the authors to put the versions of the used R packages and R version and other used software.

Reviewer #2: The manuscript by Delaluna and colleagues describes mitochondrial genome of Trichuris trichiura from Korean individuals using nanopore-based long-read sequencing. Overall, the article is well written and the findings are of interest to scientists interested in evolutionary aspects and systematics of this and closely related moths. I do find the experimental approach overall adequate; however, one major concern relates to the novelty of this work. Therefore, the impact of paper is not so that it can be published in this high-quality journal. I strongly recommend authors to find an alternative journal for their work.

PLOS authors have the option to publish the peer review history of their article (what does this mean?). If published, this will include your full peer review and any attached files.

Reviewer #1: Yes: Mohamed Hawash

Reviewer #2: No

Figure Files:

Data Requirements:

Reproducibility:

References

---

## [Editor Report · Decision Letter 1]

11 Aug 2023

Dear Dr. Song,

We are pleased to inform you that your manuscript 'De novo  assembled mitogenome analysis of Trichuris trichiura from Korean individuals using nanopore-based long-read sequencing technology' has been provisionally accepted for publication in PLOS Neglected Tropical Diseases.

Best regards,

Bruce A. Rosa

Academic Editor

Uriel Koziol

Section Editor

Thank you for addressing the reviewer concerns, and revising the manuscript accordingly.

---

## [Editor Report · Acceptance letter]

21 Aug 2023

Dear Dr. Song,

We are delighted to inform you that your manuscript, "De novo  assembled mitogenome analysis of Trichuris trichiura from Korean individuals using nanopore-based long-read sequencing technology," has been formally accepted for publication in PLOS Neglected Tropical Diseases.

Best regards,

Shaden Kamhawi

co-Editor-in-Chief

Paul Brindley

co-Editor-in-Chief
